# Selective Engineering for Preparing Entangled Steady States in Cavity QED Setup

**Emilio H. S. Sousa** 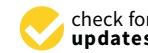 **and J. A. Roversi \***

Instituto de Fisica "Gleb Wataghin", Universidade Estadual de Campinas, 13083970 Campinas, SP, Brazil
* Correspondence: roversi@ifi.unicamp.br

**Abstract:** We propose a dissipative scheme to prepare maximally entangled steady states in cavity QED setup, consisting of two two-level atoms interacting with the two counter-propagating whispering-gallery modes (WGMs) of a microtoroidal resonator. Using spontaneous emission and cavity decay as the dissipative quantum dynamical source, we show that the steady state of this system can be steered into a two-atom single state as well as into a two-mode single state. We probed the compound system with weak field coupled to the system via a tapered fiber waveguide, finding it is possible to determine whether the two atoms or two modes are driven to a maximally entangled state. Through the transmission and reflection measurements, without disturbing the atomic state, when the cavity modes are being driven, or without disturbing the cavity field state, when a single atom being driven, one can get the information about the maximal entanglement. We also investigated for both subsystem, two-atom and two-mode states, the entanglement generation and under what conditions one can transfer entanglement from one subsystem to the other. Our scheme can be selectively used to prepare both maximally entangled atomic state as well as maximally entangled cavity-modes state, providing an efficient method for quantum information processing.

**Keywords:** entangled states; two atoms; two-modes; cavity QED setup

## 1. Introduction

The preparation and control of entangled states via dissipative engineering have attracted great interest in the last several years [1,2]. Different from the unitary evolution based schemes, these schemes use decoherence as a powerful resource in the state preparation process without destroying the quantum entanglement. In [3], the authors showed that the cavity decay plays an integral part in preparing a maximally entangled state of two Λ atoms trapped in an optical cavity. Stannigel et al. showed that the driven-dissipative preparation of entangled states can be obtained in cascaded quantum-optical networks between individual nodes [4]. In [5], the authors used the energy relaxation of the single superconducting qubit coupled to two spatially separated transmission line resonators for generating a two-mode entangled state. In addition, the generation of a two-mode entangled states has been investigated by quantum reservoir engineering [6]. Other schemes based on the waveguide QED configuration are presented in [7–12].

On the other hand, the identification of a quantum state is usually achieved by quantum state tomography [13]. However, this method directly performs a series of projective measurements on many identical copies of the quantum state, inevitably disturbing the state of the system. To circumvent this problem, quantum nondemolition measurements [14] are projected to prevent the back action of the measurement on the detected observable. Recently, proposals to realize a quantum nondemolition measurements of a superconducting flux qubit [15], pair of atomic samples [16] and nonclassical state of a massive object [17] have been demonstrated. However, an interesting question is whether we

can use an experimentally feasible method to detect an entangled state of two atoms or two modes, selectively, in a single quantum system via quantum nondemolition measurements.

Here, we report a dissipative scheme to prepare maximally entangled steady states in cavity QED setup, which consists of two two-level atoms interacting with the two counter-propagating whispering-gallery modes (WGMs) of a microtoroidal resonator. The steady state of this system can be steered into an entangled atomic state or into an entangled field state by the dissipative quantum dynamical process. In this scheme, both atomic spontaneous emission and cavity decay are utilized as a resource to engineer the targeted state. In addition, the dissipative steady-state production requires neither precise time control nor initial state preparation. To probe these entangled states without disturbing them, we performed transmission and reflection measurements through the incident weak field of two ways: (i) driving the cavity mode; and (ii) driving a single atom. For this propose, both the microtoroidal cavity and the atom were coupled to a tapered fiber waveguide and detectors. By injecting and controlling a weak field into a tapered fiber, we could determine whether the two atoms (driving the cavity mode) or two modes (driving a single atom) are in a maximally entangled state by a single click on the detector, without disturbing the atomic state or cavity field state. Thus, one of the detectors acted as a witness of the preparation of the entangled atomic state and the other of the entangled field state. Note that our goal was not to find a way to simultaneously prepare two entangled states, but to explore the possibility to get a maximally entangled state between two atoms or two modes, under certain conditions. Compared with previous proposals, the present scheme indicates a possibility of preparing an entangled atomic state as well as an entangled field state [18] as well as using the dissipation as a powerful resource to engineer in those states [19]. We also investigated the time evolution of the entanglement for both subsystem, i.e. two atoms and two modes, and under what conditions one can transfer an entangled state of two qubits from one subsystem to the other.

## 2. Model

Our system consists of a pair of identical two-level atoms that interact with the evanescent fields of a microtoroidal cavity, as shown in Figure 1. We denote the atoms by label $i = 1, 2$ with frequency $\omega_{eg}$ and the atomic ground and excited states by $|g\rangle_i$ and $|e\rangle_i$, respectively. The cavity supports two WGMs at frequency $\omega_c$ and with annihilation (creation) operators $\hat{a}$ $(\hat{a}^\dagger)$ and $\hat{b}$ $(\hat{b}^\dagger)$. These two modes have an intrinsic loss rate $\kappa_{in}$ and are coupled to each other with coupling strength $J$. Each atom is coupled simultaneously with the two WGMs via evanescent field with a coherent coupling strength described by $g_i$. dipole–dipole interaction of strength $\Omega$ is also included. The Hamiltonian of the whole system can be written in the form (in unit $\hbar$) [20–22]:

$$
\begin{aligned}
H =\ & \omega_{eg}(\hat{\sigma}_1^+\hat{\sigma}_1^- + \hat{\sigma}_2^+\hat{\sigma}_2^-) + \omega_c(\hat{a}^\dagger\hat{a} + \hat{b}^\dagger\hat{b}) + J(\hat{a}^\dagger\hat{b} + \hat{a}\hat{b}^\dagger) + \Omega(\hat{\sigma}_1^+\hat{\sigma}_2^- + \hat{\sigma}_2^+\hat{\sigma}_1^-) \\
& (g_1^*\hat{a}^\dagger\hat{\sigma}_1^- + g_1\hat{a}\hat{\sigma}_1^\dagger) + (g_1\hat{b}^\dagger\hat{\sigma}_1^- + g_1^*\hat{b}\hat{\sigma}_1^\dagger) + (g_2^*\hat{a}^\dagger\hat{\sigma}_2^- + g_2\hat{a}\hat{\sigma}_2^\dagger) + (g_2\hat{b}^\dagger\hat{\sigma}_2^- + g_2^*\hat{b}\hat{\sigma}_2^\dagger)
\end{aligned}
\tag{1}
$$

where $\hat{\sigma}_i^+ = |e\rangle_i\langle g|$ and $\hat{\sigma}_i^- = |g\rangle_i\langle e|$ are the raising and lowering operators of the atom $i$.

Introducing dissipation, the dynamics of the system is governed by the master equation [21,22]:

$$
\dot{\rho}(t) = -i[H, \rho(t)] + (\gamma/2)\sum_{i=1}^{2}\mathcal{D}[\hat{\sigma}_-^i]\rho(t) + \kappa\mathcal{D}[\hat{a}]\rho(t) + \kappa\mathcal{D}[\hat{b}]\rho(t),
\tag{2}
$$

where $\gamma$ is the spontaneous emission rate of the atoms, $\kappa$ is the cavity decay rate and $\mathcal{D}[\mathcal{O}]\rho(t) \equiv 2\mathcal{O}\rho(t)\mathcal{O}^\dagger - \mathcal{O}^\dagger\mathcal{O}\rho(t) - \rho(t)\mathcal{O}^\dagger\mathcal{O}$. The spectrum of the system, i.e., its allowed states, are represented by the eigenvalues and eigenvectors of $H$, in the ideal case (note that, including system dissipation, the eigenvectors with zero eigenvalues are still the same stationary entangled states present in the ideal case, whereas the others two decay in time), are given by

$$
E_0^0 = 0; \qquad |E_0^0\rangle = |gg00\rangle
$$

$$E_1^1 = 0; \qquad |E_1^1\rangle = \frac{1}{\sqrt{g_1^2 + g_2^2}} \left( g_1 |eg00\rangle - g_2 |ge00\rangle \right)$$

$$E_1^2 = 0; \qquad |E_1^2\rangle = \frac{1}{\sqrt{g_1^2 + g_2^2}} \left( g_1 |gg10\rangle - g_2 |gg01\rangle \right)$$

$$E_1^{\pm} = \pm i\sqrt{2}\sqrt{g_1^2 + g_2^2}; \qquad |E_1^{\pm}\rangle = \frac{1}{\sqrt{2}} \left\{ \frac{1}{\sqrt{2}} (|gg10\rangle + |gg01\rangle) \pm \frac{1}{\sqrt{g_1^2 + g_2^2}} (g_1 |eg00\rangle + g_2 |ge00\rangle) \right\}$$

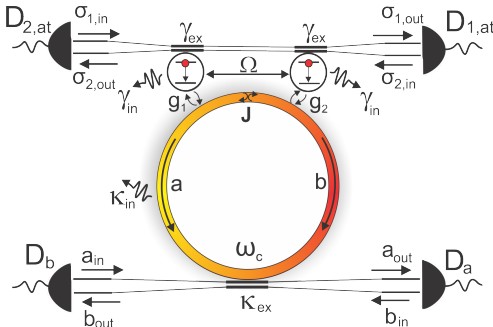

**Figure 1.** Experimental scheme of the atom–cavity system. The cavity consists of two WGMs coupled simultaneously to a pair of two-level atoms interacting via dipole–dipole interaction $\Omega$. In this case, both the cavity and the atom are coupled to a tapered optical fiber in an overcoupled regime [21]. The modes of fiber are described by $\{\hat{a}_{in}, \hat{a}_{out}, \hat{b}_{in}, \hat{b}_{out}\}$ coupled to the cavity and $\{\hat{\sigma}_{1,in}, \hat{\sigma}_{1,out}, \hat{\sigma}_{2,in}, \hat{\sigma}_{2,out}\}$ coupled to the atom, in terms of the input–output fields to a detectors.

Note that, for $g_1 = g_2$, the eigenvector correspondent to the eigenvalue $E_1^1 = 0$ is a tensorial product between the maximally entangled states of the atoms and the cavity in a vacuum state and for $E_1^2 = 0$ is a tensorial product between the atoms in ground state and a maximally entangled states between the field modes,

$$|E_1^1\rangle_{g_1=g_2} = \frac{1}{\sqrt{2}}(|eg\rangle - |ge\rangle) \otimes |00\rangle = |\psi^-, V\rangle$$

$$|E_1^2\rangle_{g_1=g_2} = |gg\rangle \otimes \frac{1}{\sqrt{2}}(|10\rangle - |01\rangle) = |G, \phi^-\rangle$$

where $|G\rangle = |gg\rangle$, $|V\rangle = |00\rangle$, $|\psi^-\rangle = \frac{1}{\sqrt{2}}(|eg\rangle - |ge\rangle)$ and $|\phi^-\rangle = \frac{1}{\sqrt{2}}(|10\rangle - |01\rangle)$. The energy-level diagram of the atom–toroid system with the transitions and decay rates, via Fermi's golden rule (without spontaneous emission ($\gamma = 0$), collective decay rates of the modes are given by $\Gamma_c = \kappa/2$ e $\Gamma_c^{\pm} = \kappa$, while, without cavity loss ($\kappa = 0$), the collective decay rates of the atoms are given by $\Gamma_a = \gamma/4$ e $\Gamma_a^{\pm} = \gamma/2$ [23], and the probe fields (which are discussed further below) are shown in Figure 2. In this study, we only considered the lowest energy of the system, since we were interested in its steady state, which is a mixture of these states for any initial state of the system. Thus, the steady state of the system is a mixture of the lower energy states of each subspace:

$$\rho_{ss} = (1 - P_{spont} - P_{cav})|G,00\rangle\langle G,00| + P_{spont}|\phi^-,00\rangle\langle\phi^-,00| + P_{cav}|\psi^-,00\rangle\langle\psi^-,00| \tag{3}$$

where $P_{spont}$ is the projection of the initial state of the atom in the state $|\psi^-\rangle$ and $P_{cav}$ is the projection of the initial state of the fields in the state $|\phi^-\rangle$. Equation (3) can be obtained directly from Equation (2) to $t \to 0$.

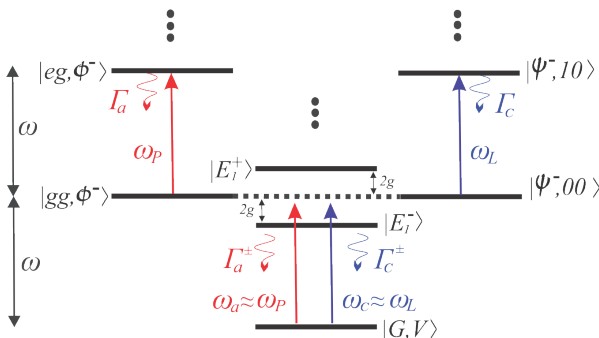

**Figure 2.** Energy levels diagram of the atom-cavity system with collective decay rates and probe fields where (blue) driving the cavity mode with strength $\epsilon$ and frequency $\omega_L$ and (red) driving the single atom with strength $\eta$ and frequency $\omega_P$.

Following the method described in [18], to discriminate these states without disturbing them, we must monitor the system using a weak probe field, keeping the system still with a single excitation. In our case, we used two distinct procedures to monitor the atom–cavity system via probe field: (i) drive the cavity mode to distinguish between the atomic states $|G\rangle$ and $|\psi^-\rangle$, restricted to the time interval $\kappa/g^2 \ll t < 1/\gamma$ [24]; and (ii) drive a single atom to distinguish between the states of the fields $|V\rangle$ and $|\phi-\rangle$ in the time interval $\kappa t \ll 1$ and $g^2 t/\gamma \gg 1$. In both cases, such discrimination of states consisted in measuring the coefficients of transmission and reflection of the incident field, without disturbing the atomic system (Case *i*) or the cavity modes (Case *ii*). In addition, using the formalism of input–output theory [25], the output fields are given by

$$a_{out}(t) = a_{in}(t) + \sqrt{2\kappa_{ex}}a(t) \tag{4}$$

$$b_{out}(t) = b_{in}(t) + \sqrt{2\kappa_{ex}}b(t) \tag{5}$$

where the coherent amplitudes of the input fields are given by $\langle a_{in}\rangle = \frac{i\epsilon}{\sqrt{2\kappa_{ex}}}$ and $\langle b_{in}\rangle = 0$ [22]. In this way, the transmission and reflection coefficients of the modes are defined as

$$T_c = \frac{\langle a_{out}^\dagger a_{out}\rangle_{ss}}{|\epsilon|^2/(2\kappa_{ex})} \tag{6}$$

$$R_c = \frac{\langle b_{out}^\dagger b_{out}\rangle_{ss}}{|\epsilon|^2/(2\kappa_{ex})}. \tag{7}$$

Similarly, the transmission and reflection coefficients from the atoms are given by

$$T_a = \frac{\langle \sigma_{1,out}^+\sigma_{1,out}^-\rangle_{ss}}{|\eta|^2/(2\gamma_{ex})} \tag{8}$$

$$R_a = \frac{\langle \sigma_{2,out}^+\sigma_{2,out}^-\rangle_{ss}}{|\eta|^2/(2\gamma_{ex})}. \tag{9}$$

## 3. Monitoring the Atom–Cavity System by Driven the Cavity Mode

Our purpose in this section is to probe stationary atomic states, that is, to distinguish an uncorrelated atomic state ($|G\rangle$) from a maximally entangled state ($|\psi^-\rangle$) without disturbing them, e.g., by performing a quantum nondemolition measurement. In this case, the mode $a$ of the cavity is driven by a probe field given by $H_L = \epsilon(\hat{a}e^{i\omega_L t} + \hat{a}^\dagger e^{-i\omega_L t})$, where $\epsilon$ and $\omega_L$ are the strength and the frequency of the probe field, respectively. In Figure 2, we can observe that, if the system is in the state $|\psi-,00\rangle$, the external field is capable of promoting the transition $|\psi^-,00\rangle \to |\psi^-,10\rangle$, resonantly. Note that $|\psi^-\rangle$ is a dark state and, in this case, the atoms do not "see" the cavity modes and this would be the same as if imposing $g \simeq 0$, thus, obtaining $T_c = 1$ and $R_c = 0$. On the other hand, if the system

is in the state $|G, V\rangle$, the external field leads to the transition $|G, V\rangle \rightarrow |E_1^\pm\rangle$ having detuning $(\Delta_L)$ between the frequencies of the probe and atom–field system given by $\pm 2g$, resulting in $T_c = 0$ and $R_c = 1$, as explained below. This occurs because the two atoms are strongly coupled to the cavity, i.e., $|g| \gg \{\kappa_{in}, J, \Omega, |\Delta_L|, \Gamma_c\}$, resulting in an intracavity field redistribution between counter-propagating modes $a$ and $b$ [21]. Thus, the radiated field is dynamically controlled by atomic polarization which produces a destructive interference in the output field $a_{out}$ between the components $a_{in}$ and $\sqrt{2}\kappa_{ex}$ resulting in a transmission $T_c \rightarrow 0$, and, consequently, the incident field is fully reflected in the fiber via the output field $b_{out}$, resulting in $R_c \rightarrow 1$ (note that $\mathcal{T}_c + \mathcal{R}_c < 1$ due to the losses in the system) [21,26]. Therefore, the transmission and reflection of the probe field applied to the cavity mode can be used as witness of the atomic states, without disturbing them so that, $T_c = 1$ and $R_c = 0$ implies that $\rho_{ss}^{at} \rightarrow |\psi^-\rangle\langle\psi^-|$, and $T_c = 0$ and $R_c = 1$ implies that $\rho_{ss}^{at} \rightarrow |G\rangle\langle G|$. In a compact way, the steady state of the atomic system, tracing over the modes of the cavity, is given by:

$$Tr_c[\rho_{ss}] \rightarrow \rho_{ss}^{at} = (1 - P_{spont})|G\rangle\langle G| + P_{spont}|\psi^-\rangle\langle\psi^-|. \tag{10}$$

Figure 3 shows the transmission and reflection of the cavity modes as a function of the detuning $(\Delta_L = \omega_c - \omega_L)$ between the frequencies of the probe field and the atom–toroid system in overcoupled regime, for $\{\epsilon, \kappa_{ex}, \kappa_{in}, \Omega, J, \Gamma_c, g\}/2\pi = \{10, 20, 0.2, 0, 0, 5.2, 45\}$MHz. In Figure 3, the dashed line correspond to the case where $\rho_{ss}^{at} \rightarrow |\psi^-\rangle\langle\psi^-|$, so that the incident field is fully transmitted ($T_c = 1$ and $R_c = 0$), equivalent to empty cavity type behavior ($g = 0$). On the other hand, when $\rho_{ss}^{at} \rightarrow |G\rangle\langle G|$ (solid line), it was observed that, near the resonance ($\Delta_L = 0$), the strong coupling between the atoms and the cavity modes, that is, $g > \kappa_{ex}$, causes the incident field to be fully reflected ($T_c = 0$ and $R_c = 1$) and, in this case, the atoms act as a "mirror" for the incident field.

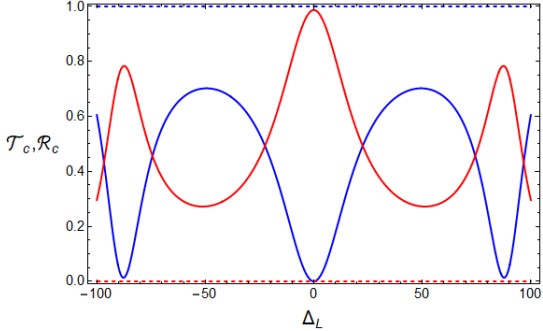

**Figure 3.** Transmission (blue) and reflection (red) of the cavity modes as a function of the detuning between the frequencies of the probe field and the atom–cavity system in overcoupled regime $\kappa_{ex} \gg \{\kappa_{in}, J\}$ and $\omega_c = \omega_a$. The chosen parameters were $\{\epsilon, \kappa_{ex}, \kappa_{in}, \Omega, J, \Gamma_c, g\}/2\pi = \{10, 20, 0.2, 0, 0, 5.2, 45\}$ MHz. The dashed line corresponds to $\rho_{ss}^{at} \rightarrow |\psi^-\rangle\langle\psi^-|$ and the solid line for $\rho_{ss}^{at} \rightarrow |G\rangle\langle G|$.

## 4. Monitoring the Atom–Cavity System by Driven the Single Atom

In this section, we probe the steady states of cavity modes, that is, differentiate an uncorrelated state ($|V\rangle$) from an entangled state between two modes ($|\phi\rangle$) by driving the single atom, without disturbing the states of the cavity modes. In this setup, the incident field is given by $H_P = \eta(\sigma_1^- e^{i\omega_P t} + \sigma_1^+ e^{-i\omega_P t})$, where $\eta$ and $\omega_P$ are the strength and the frequency of the probe field, respectively. Note that the external field was applied to atom 1, as shown in Figure 2. It was also observed that the probe field is capable of promoting, resonantly, the transition $|gg, \phi^-\rangle \rightarrow |eg, \phi^-\rangle$. Similar to the previous discussion, $|\phi^-\rangle$ is a dark state and, in this case, the cavity modes do not "see" atoms and this would be the same as if imposing $g \simeq 0$, resulting in $T_a = 1$ and $R_a = 0$. However, for $|gg, V\rangle$, the incident field is capable of promoting the transition, of resonance, $|gg, V\rangle \rightarrow |E_\pm\rangle$ with detuning $\pm 2g$, resulting in $T_a = 0$ and $R_a = 1$. In this case, the incident field is fully reflected due to destructive interference between the components $\sigma_{1,in}^-$ and $\sqrt{2\gamma_{ex}}\sigma_1^-$ resulting in $T_a \rightarrow 0$.

Therefore, the transmission and reflection of the probe field applied on atom 1 can be used as witness of the state of the cavity modes without disturbing them, so that, $T_a = 1$ and $R_a = 0$ implies that $\rho_{ss}^c \rightarrow |\phi^-\rangle\langle\phi^-|$ and $T_a = 0$ and $R_a = 1$ implies that $\rho_{ss}^c \rightarrow |V\rangle\langle V|$. The steady state of the cavity modes, tracing over the atomic states, is given by:

$$Tr_{at}[\rho_{ss}] \rightarrow \rho_{ss}^c = (1 - P_{cav})|V\rangle\langle V| + P_{cav}|\phi^-\rangle\langle\phi^-|. \tag{11}$$

Figure 4 shows the transmission and reflection of the atoms as a function of the detuning between the frequencies of the probe field and the atom–toroid system $\Delta_P = \omega_a - \omega_P$ in an overcoupled regime, for $\{\eta, \gamma_{ex}, \gamma_{in}, \Omega, J, \Gamma_a, g\}/2\pi = \{10, 40, 0.2, 0, 0, 5.2, 45\}$ MHz.

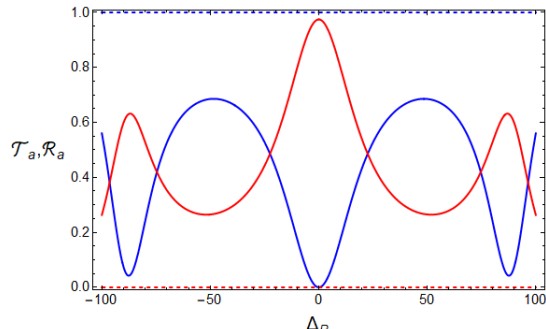

**Figure 4.** Transmission (blue) and reflection (red) of the atoms as a function of the detuning between the frequencies of the probe field and the atom–cavity system in an overcoupled regime $\gamma_{ex} \gg \{\gamma_{in}, \Omega\}$ and $\omega_c = \omega_a$. The chosen parameters were $\{\gamma_{ex}, \gamma_{in}, \Omega, J, \Gamma_c\}/2\pi = \{40, 0.2, 0, 0, 5.2\}$ MHz. The dashed line corresponds to $\rho_{ss}^c \rightarrow |\phi^-\rangle\langle\phi^-|$ and the solid line for $\rho_{ss}^c \rightarrow |V\rangle\langle V|$.

In Figure 4, the dashed lines correspond to the case that $\rho_{ss}^c \rightarrow |\phi-\rangle\langle\phi-|$, where the probe field is fully transmitted ($T_a = 1$ and $R_a = 0$), because, in this state, the cavity modes do not "see" atoms ($g = 0$). However, when $\rho_{ss}^c \rightarrow |V\rangle\langle V|$ (solid lines), it is noted that, around the resonance ($\Delta_P = 0$), the strong coupling between the modes and the atoms ($g > \gamma_{ex}$) induces a complete reflection of the incident field ($T_a = 0$ and $R_a = 1$).

## 5. Transfer of Entanglement between Two Atoms and Two Modes

In this section, we investigate the case when the system is initially prepared in an entangled state and under what conditions it is possible to obtain the transfer of entanglement between the two subsystems, e.g., from the two atoms to the two modes. In this case, we consider the following two initial states: $|\phi(0)\rangle_1 = \frac{1}{\sqrt{2}}(|eg\rangle + |ge\rangle) \otimes |00\rangle$ and $|\phi(0)\rangle_2 = \frac{1}{\sqrt{2}}(|10\rangle + |01\rangle) \otimes |gg\rangle$. For this purpose, we use negativity [27] as measure of the degree of entanglement between the two atoms ($\mathcal{N}_a$) and the two-modes ($\mathcal{N}_c$).

Figure 5 shows the time evolution of the negativity between the atoms (blue lines) and between the fields (red lines) as a function of the scaled time $gt$ for the initial states $|\phi(0)\rangle_1$ (left) and $|\phi(0)\rangle_2$ (right) with $\gamma$ fixed and different values of $\kappa$. It was observed that, in both initial states, it is possible to completely transfer the entanglement between the two subsystems when $\kappa = \gamma = 0.01g$ (next to the ideal case), as expected. For small values of $gt$ and higher values of $\kappa$, the degradation of the entanglement between atoms is more robust when compared to the entanglement between the modes.

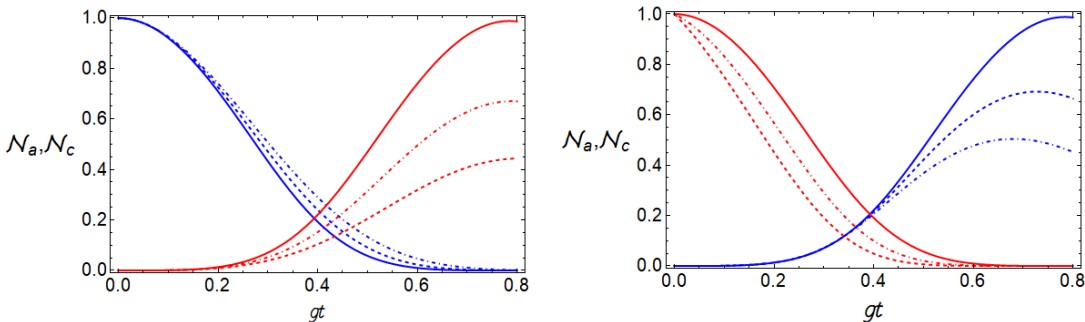

**Figure 5.** Time evolution of the negativity between the atoms (blue) and fields (red) as a function of time scaled $gt$ for the initial states: (**left**) $|\phi(0)\rangle_1 = \frac{1}{\sqrt{2}}(|eg00\rangle + |ge00\rangle)$ ; and (**right**) $|\phi(0)\rangle_2 = \frac{1}{\sqrt{2}}(|gg10\rangle + |gg01\rangle)$. The chosen parameters were $\gamma = 0.01g$ and $\Omega = J = 0$ for $\kappa = 0.01g$ (solid line), $\kappa = 0.5g$ (dot-dashed line) and $\kappa = 1.0g$ (dashed line).

## 6. Conclusions

In conclusion, using a dissipative atom-microtorid system, we have shown that it is possible to prepare a selectively maximally entangled state between two atoms as well as a maximally entangled state between two modes. In this case, the results were obtained by measuring the transmission and reflection coefficients of a weak probe field applied to the atom–cavity system, so as not to disturb the atomic state (drive the cavity mode) or the state of the cavity field (drive the single atom) by performing a quantum nondemolition measurement. In addition, we have shown that it is possible to transfer an entangled state of two qubits from the two atoms to the two modes under certain conditions. Therefore, our results may contribute to a better understanding of the preparation and transfer of entangled states that are of great interest to the quantum information processing.

**Author Contributions:** All authors contributed substantially to the research.

**Funding:** This research was funded by Conselho Nacional de Desenvolvimento Científico e Tecnológico (CNPq) grant number 140039/2016-3 (http://www.carloschargas.cnpq.br), Coordenação de Aperfeiçoamento de Pessoal de Nível Superior (CAPES) grant number 1247693/2013 (https://www.capes.gov.br), Fundação de Apoio a Pesquisa do Estado de São Paulo (FAPESP) (http://www.fapesp.br) and Instituto Nacional de Ciência e Tecnologia (INCT) (http://www.inct-iq@if.ufrj.br).

**Acknowledgments:** The authors would like to thank Mohinder Paul Sharma for the valuable comments and careful English revision of the manuscript.

**Conflicts of Interest:** The authors declare no conflict of interest.

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
