# Peer review of "Selective Engineering for Preparing Entangled Steady States in Cavity QED Setup"

_quantumrep, doi:10.3390/quantum1010007_

Round 1
Reviewer 1 Report
Referee Report quantumrep-529311 (June 2019)
PAPER DESCRIPTION
In the paper titled “Selective engineering for preparing entangled steady states in circuit QED setups” Sousa et al, have studied atomic and cavity mode entanglement generation and time evolution in two-atom -cavity QED setup. The authors proposed the generation is verified through a quantum non-demolition measurement by introducing a weak field through the waveguide and recording its transmission and reflection spectrum. Robustness of the proposed scheme has been discussed.
DECISION
Entanglement generation in quantum optical driven-dissipative systems lies at the heart of several protocols in quantum information science. In particular, the setup studied by the authors (cavity QED) is a rich architecture to host such quantum effects in a controlled way. In this regard, the authors have investigated an important subject which is also timely from an experimental point of view. Moreover, the paper is systematically presented. However, there are some points which in my opinion need further attention. Therefore, I can recommend publication of this manuscript, provided the author performs the following revisions:
COMMENTS AND QUESTIONS
• Is the spectrum of the system as mentioned below Eq. 2 works for when dissipations are also included or its just for closed system dynamics? Please specify.
• Please discuss the validity of Eq. 3, meaning how realistic under actual experimental conditions is to achieve such a density operator?
• The entanglement generation and control in waveguide QED and cavity QED setups have seen many interesting findings in the recent past. To put this work in connection with some of the recent works, I would suggest authors cite some of the recent papers on this subject for instance:
I. Physical Review A, 99, 042315 (2019) (“Accelerating adiabatic protocols for entangling two qubits in circuit QED “).
II. Physical Review A, 94, 012309 (2016) (“Two-photon entanglement in multiqubit bidirectional-waveguide QED “).
III. Physical Review A, 98, 062327 (2018) (“Simple preparation of Bell and Greenberger-Horne-Zeilinger states using ultrastrong-coupling circuit QED “).
IV. Physical Review A, 94, 012302 (2016) (“Multiqubit entanglement in bidirectional-chiral-waveguide QED “).
V. Journal of Modern Optics, 62, 13, 1048-1060 (2015) (“Controlling tripartite entanglement among optical cavities by reservoir engineering”).
VI. Physics Letters A, 379, 28-29, 1643-1648 (2015) (“Bi- and uni-photon entanglement in two-way cascaded fiber-coupled atom-cavity systems).
• In the conclusion authors have mentioned “circuit QED” but in the actual manuscript, there is no discussion related to circuit QED. The starting Hamiltonian and the calculations presented are generally true for any cavity QED platform. I am unable to see why authors are specifying circuit QED setups here. Please briefly discuss in the beginning.
TYPOS
I have noticed several typos in the present manuscript. Please read the paper carefully to remove these typos. Some of the typos I am pointing out in the following:
• Abstract, the ninth and tenth line from the top: Please correct “been driving” to “being driven”.
• Authors have used “probe field” and “prove field”. Please clearly state what is the difference between the two. If this is a typo please correct.
Reviewer 2 Report
The manuscript describes quantum state engineering with two
two-level atoms interacting with two counter-propagating
whispering-gallery modes of a microtoroidal resonator. The main
purpose of this engineering is to generate maximally entangled
atomic and entangled cavity-mode states.
I find the topic and some of the obtained results interesting and
worth of publication. I can recommend the publication of this work
in Quantum Reports if the authors adequately revise the manuscript
to address the following comments:
(1) The title stresses that the paper describes a "circuit QED
setup" but the paper does not contain any details related sensu
stricto to circuit QED. One could automatically replace the term
"circuit QED" by "cavity QED" in the entire file without any
problem.
Indeed, the abstract explains that "circuit QED setup, which
consists of two two-level atoms interacting with the two
counter-propagating whispering-gallery modes (WGMs) of a
microtoroidal resonator." Figure 1 explains more details about the
setup, without however any details related to circuit QED.
In typical circuit QED systems, *microwave* photons are coupled to
*superconducting* qubits acting as *artificial* atoms. But the
manuscript does not mention such keywords at all.
I would suggest the Authors to give more details about their
"circuit QED setup". I mean to specify that waveguide microwave
resonators and artificial atoms (e.g., superconducting quantum
circuits like transmons, etc.) typically used in circuit QED. A
comprehensive recent review on circuit QED might be:
X. Gu et a., Physics Reports 718–719 (2017) 1–102.
Note that microtoroidal resonators with a quantized
whispering-gallery modes were first introduced in ref. [8] in a
*cavity*-QED setup.
(2) The first sentence of the Introduction reads:
"The preparation and *monitorization* of entangled states via
dissipative engineering have attracted *great* interest in the
*last* years [1,2]."
[1] Lin, Y.; ... Nature 2013.
[2] Krauter, H.; ... Phys. Rev. Lett. 2011.
I am surprised that only these two old (and not the first papers
in the field) are cited to show this "*great* interest in the *last* years".
I would suggest to cite a review or more recent papers on
dissipative quantum state engineering.
Moreover, I would replace the term "monitorization" by "control".
(3) The eight terms in the second line of Eq. (1) can be written
more compactly as, e.g.,
$\sum_{i=1,2} g_i(\hat a+\hat b)\hat \sigma_i^+ +h.c.$
(4) Does it hold $T_c+R_c=1$? If yes, this should be mentioned. If
not, this should be explained.
(5) Figs. 3 and 4 look very similar, so maybe better to show
either all these curves in one figure or the two transmission curves
in one figure and the two reflection curves in another figure.
Minor comments:
* $\Psi^-$ and $\Phi^_$ should be replaced by $\psi^-$ and $\phi^_$
in Fig. 2 to be consistent with the symbols used in, e.g., Eq. (3).
* In the caption of Fig. 2, please write explicitly
the mathematical symbols for the "prove fields" in Fig. 2.
What is the meaning of the "prove fields": "test fields" or "probe fields"?
* Better to display as numbered equations those formulas in the text
just before Sec. 3, i.e.: (1) the input-output relations, and
(2,3) the transmission and reflections coefficients.
* If we write hats in $\hat a$ and $\hat b$, these should be used
systematically over all operators, including \sigma operators.
Hats are missing in $\hat a$ and $\hat b$ in the caption of Fig.~1.
* The term "Fermi's gold rule" should read "Fermi's golden rule".
Round 2
Reviewer 2 Report
The Authors have revised their paper to address the main critical
comments of my first report. In particular, they have modified the
title, abstract, and main text of their paper by replacing the
term "circuit QED" by "cavity QED". Thus, I can recommend this
revised paper for publication in Quantum Reports.